# Species-Specific Responses to Human Trampling Indicate Alpine Plant Size Is More Sensitive than Reproduction to Disturbance

**DOI:** 10.3390/plants12173040

**Published:** 2023-08-24

**Authors:** Nathalie Isabelle Chardon, Philippa Stone, Carly Hilbert, Teagan Maclachlan, Brianna Ragsdale, Allen Zhao, Katie Goodwin, Courtney G. Collins, Nina Hewitt, Cassandra Elphinstone

**Affiliations:** 1Biodiversity Research Centre, University of British Columbia, Unceded x^w^məθk^w^əy’əm (Musqueam) Territory, 2212 Main Mall, Vancouver, BC V6T 1Z4, Canada; philippa.stone@botany.ubc.ca (P.S.); katiegoodwin321@gmail.com (K.G.); courtney.collins@botany.ubc.ca (C.G.C.); cassandra.elphinstone@shaw.ca (C.E.); 2Department of Botany, University of British Columbia, Unceded x^w^məθk^w^əy’əm (Musqueam) Territory, 3156-6270 University Blvd., Vancouver, BC V6T 1Z4, Canada; 3University of British Columbia, Unceded x^w^məθk^w^əy’əm (Musqueam) Territory, Vancouver, BC V6T 1Z4, Canada; carlyhilbert02@gmail.com (C.H.); teagmaclachlan@gmail.com (T.M.); bri.ragsdale15@gmail.com (B.R.); allen10to11@gmail.com (A.Z.); 4Department of Geography, University of British Columbia, Unceded x^w^məθk^w^əy’əm (Musqueam) Territory, Vancouver, BC V6T 1Z4, Canada; nina.hewitt@ubc.ca

**Keywords:** alpine plants, human trampling disturbance, global change, recreation, plant traits, trail planning

## Abstract

Human disturbance, such as trampling, is an integral component of global change, yet we lack a comprehensive understanding of its effects on alpine ecosystems. Many alpine systems are seeing a rapid increase in recreation and in understudied regions, such as the Coast Mountains of British Columbia, yet disturbance impacts on alpine plants remain unclear. We surveyed disturbed (trail-side) and undisturbed (off-trail) transects along elevational gradients of popular hiking trails in the T’ak’t’ak’múy’in tl’a In’inyáxa7n region (Garibaldi Provincial Park), Canada, focusing on dominant shrubs (*Phyllodoce empetriformis, Cassiope mertensiana*, *Vaccinium ovalifolium*) and graminoids (*Carex* spp). We used a hierarchical Bayesian framework to test for disturbance by elevation effects on total plant percent cover, maximum plant height and diameter (growth proxies), and buds, flowers, and fruits (reproduction proxies). We found that trampling reduces plant cover and impacts all species, but that effects vary by species and trait, and disturbance effects only vary with elevation for one species’ trait. Growth traits are more sensitive to trampling than reproductive traits, which may lead to differential impacts on population persistence and species-level fitness outcomes. Our study highlights that disturbance responses are species-specific, and this knowledge can help land managers minimize disturbance impacts on sensitive vegetation types.

## 1. Introduction

Subalpine and alpine plants are sensitive to disturbances from trampling due to their perennial nature, short growing seasons, and shallow soils [1,2]. Species’ responses to trampling vary, with different effects seen in growth and reproduction. These species-specific responses may be related to differences in life history. Grime [3] defined combinations of different life history traits that make certain groups of species more successful along gradients of disturbance, stress, and competition. Ruderals, such as grasses and sedges, are most common in areas with high disturbance, whereas slower-growing, competitive species do not tolerate disturbance well. For example, studies have found that *Phyllodoce* spp. (slow-growing subalpine heathers) are vulnerable to trampling disturbances with reduced heights, whereas *Carex* spp. (sedges) are more resilient [4,5]. The effects of trampling on soils can even have long-term effects on plant community succession, with greater diversity at trampled sites in the Cascade Mountains in Washington [6].

In addition to reducing plant size, trampling can also reduce reproductive output, interrupt succession, and lower species richness [7,8,9,10,11,12,13,14,15]. Disturbance can also decrease total plant cover, but the severity of effects differs between ecosystems [14,16,17]. Reduced size, reproduction, and survival due to trampling can scale up to negatively affect overall population performance and potentially reduce long-term population viability, especially for hiking trails within small, endangered community types [18]. Population performance may be less sensitive to trampling when this disturbance affects reproduction more than survival [19], and thus the plant tissue that is most affected by trampling will likely influence how trampling impacts overall population performance. However, the extent that trampling might differently affect vegetative growth versus reproduction, and whether this effect is uniform across species, is understudied.

One of the main mechanical effects of trampling on plant habitat is that trampling disturbance induces soil compaction. Increased soil compaction has been found to reduce pore connectivity, which in turn decreases its hydro-conductivity and root permeability [20]. Soil compaction also reduces the relative abundance of soil microbes and mycorrhizal fungi [21]. Such conditions can make trampled habitats inhospitable for many plant species. Elevation may also play an important role in how plants and soils respond to disturbance [22]. Climate, including precipitation and temperature, rates of soil development and hydrology, relative plant canopy cover, and numerous other factors vary consistently across elevation gradients, and thus the response of vegetation to the same intensity of disturbance may vary greatly at different elevation zones [22,23,24]. Furthermore, species diversity in a plant community can greatly influence sensitivity to human trampling, with mixed communities of subalpine plant species having three times higher survival than the same species in pure stands [25].

The synergistic effects of disturbance and climate further complicate our understanding of trampling impacts on plant communities. Climatic conditions can have varying effects on alpine communities depending on region, and regional climate changes can negatively impact the rate of plant regeneration post-disturbance [2,26]. In other cases, increased precipitation and warmer summers may even increase the richness of alpine plant species in general, although this response is individualistic and regional [27]. These changes could mediate or exacerbate the effects of trampling in unique ways with contrasting responses to trampling across different climatic zones [28,29]. Taken together, the complex interplays between anthropogenic disturbances and climate change emphasize the need to study these effects in alpine regions experiencing rapid changes in climate and human recreation.

One region experiencing rapid changes in climate and human recreation is British Columbia’s (BC) Coast Mountains. These ocean-proximate mountains are unique on a global scale as they are among the most southerly glaciated mountains of the northern hemisphere and receive high levels of precipitation year-round [30]. Glacial landscapes and alpine ecosystems, such as those in this region, experience greater warming than other regions in the globe and have seen increasingly rapid glacial retreat in recent years [31]. They are also near Vancouver, Canada, a major population center of 2.64 million people [32]. This large and growing population has led to a rapidly increasing number of outdoor enthusiasts and new trails are being built into the most accessible mountain terrain. For example, visits to Garibaldi Provincial Park within this region increased by 62% (3-year average) between the periods 2006–2009 and 2015–2018 [33,34,35]. Understanding how these BC ecosystems will respond to warming and increasing trampling due to recreation will help park management determine the least sensitive areas for infrastructure development. However, our understanding of the effects of human trampling disturbance on alpine plant communities in the Coast Mountains of BC is surprisingly limited.

To our knowledge, only two studies have examined the effects of trampling in BC. In a managed subalpine/montane forest stand, one study found negative effects of trampling by livestock (cattle) on lodgepole pine tree plantations primarily in the first 2–3 years of planting in the Kamloops and Merritt Forest Districts [36]. Another study found that disturbance reduces plant percent cover in montane to alpine systems, but does not affect species richness in Mount Robson Provincial Park [37]. Studies in nearby alpine ecosystems in the Colorado Rockies have documented immediate and long-term impacts of trampling disturbance, including decreasing plant cover, lower reproductive output, soil erosion, and changed plant community structure [2,29,38]. However, we expect these findings to differ from trampling responses in the BC Coast Mountains, where a much deeper winter snowpack may buffer trampling impacts through winter but relatively high precipitation and snowmelt in summer may increase mud, resulting in trail widening by hikers.

Given the lack of information on disturbance impacts in the alpine systems of the BC Coastal Mountains, here we study how human trampling disturbance affects characteristic plant species in this region. Our goal is to increase our understanding of how increased human recreation might affect alpine plant communities found across the globe and help to inform future trail designs to minimize disruption locally. We surveyed dominant and common plant communities on and off popular hiking trails in the T’ak’t’ak’múy’in tl’a In’inyáxa7n (“Landing Place of the Thunderbird”) region (Garibaldi Provincial Park, British Columbia, Canada) in the Squamish and Lil’wat territories. We quantified the impacts of trampling at community (plant percent cover) and species (growth and reproductive traits) levels over elevational gradients to ask: How does human trampling disturbance affect alpine plant communities and proxy traits for plant growth and reproduction, and does this effect vary with elevation?Which plant species and functional types (evergreen shrubs, deciduous shrubs, sedges) are most sensitive to human trampling disturbance?

We computed plant percent cover from standardized photographs and measured traits on the dominant species in the heath, heather, blueberry, and sedge meadows found frequently throughout this region, encompassing evergreen and deciduous perennial shrubs and graminoids. Our focal species represent common subalpine and alpine plant communities found across arctic and alpine regions worldwide, making our results relevant to other regions around the world. Because of their frequency and broad distribution in the study area, we selected *Phyllodoce empetriformis* (Sm.) D. Don (Ericaceae) and *Cassiope mertensiana* (Bong.) G. Don (Ericaceae)*, Vaccinium ovalifolium* Sm. (Ericaceae), and *Carex* spp. (Cyperaceae) (Figure 1). We included a berry species due to their importance to the local Sḵwx̱wú7mesh (Squamish) and Lil’wat peoples [30,39]. These species provide other important ecosystem services including forage and habitat for wildlife [40], and soil stabilization and high soil carbon storage through fungal mycelium [41]. In addition, the local park authority, BC Parks, aims to better understand how recreation affects common plant communities to help plan for needed infrastructure in areas experiencing an immense increase in recreational traffic, making our study of particular interest from a management perspective.

At a community level, we predict that disturbance will reduce plant percent cover. At a species level, we predict that graminoids (with multiple ramets) will be the least sensitive to disturbance based on their quick recolonization and high survival in disturbed areas [42]. We further predict that the deciduous (*V. ovalifolium*) or evergreen (*P. empetriformis, C. mertensiana*) brittle shrubs will be more sensitive to trampling, with the slow-growing evergreen shrubs most affected by multi-year repeated disturbances [42]. Regarding the effects of trampling on plant size, we hypothesized that trampling would break branches and thus decrease plant size. We expected slow-growing species (*P. empetriformis, C. mertensiana,* and *V. ovalifolium*) to be most sensitive to this. With regard to reproduction, we expected that trampling will either (i) decrease fruit numbers due to smaller plant biomass, but not change the relative number of reproductive structures per biomass of the plant [43], or (ii) increase the density of reproductive structures on trampled plants in response to higher stress [44]. Our results suggest that species have different sensitivities to human trampling, with *V. ovalifolium* being the most sensitive in the T’ak’t’ak’múy’in tl’a In’inyáxa7n region, and elevation might not be as important as widely understood. Future studies will be needed to determine if the responses to trampling that we observed are species-specific in this region or can be generalized for broader groups (e.g., other graminoids, evergreen shrubs, or deciduous shrubs [42]).

## 2. Results

At the community level, we found that disturbance reduced plant percent cover in quadrats located on trails compared to those off trails (Table 1, Figure 2). At the species level, we found that trampling disturbance reduced size in two out of four species and reproduction in one species, although the intensity of this effect varied by species and some species’ traits were not affected by disturbance. Disturbance increased species size in only one case. In particular, disturbance reduced both the maximum height and maximum diameter in the deciduous shrub (*V. ovalifolium*), making this the species with the highest sensitivity to disturbance. However, disturbance only reduced the maximum diameter in one evergreen shrub (*P. empetriformis*) but had no effect on the growth proxies of the other evergreen shrub (*C. mertensiana*). For the sedges (*Carex* spp.), disturbance increased maximum height and had no effect on diameter (Table 1, Figure 2).

Of the three shrub species for which we quantified relative reproductive output (defined as summed buds, flowers, and fruit per plant area, relative to maximum reproductive output for that species), disturbance only affected the reproductive output of the evergreen shrub *C. mertensiana* (Table 1, Figure 2). *Cassiope mertensiana* individuals growing on the trail had a significantly lower reproductive output (per plant area) than those growing off the trail. Individuals of *P. empetriformis* and *V. ovalifolium* showed no difference in reproductive output whether growing on or off the trail. However, total mean and median reproduction were lower on the trails, and since larger plants produce more reproductive structures (Generalized Linear Mixed Model with random effects of species and transect pair: estimate = 0.019, standard error = 0.0009, *p*-value < 0.001), and this is likely due to smaller plant sizes on trails. The effects of disturbance and elevation only interacted in one case, such that the positive effects of trampling increased with elevation on only *Carex* spp. height (Table 1). However, the effects of trampling did not change with elevation for plant percent cover or other species traits. Elevation alone, within the narrow range we studied, did not affect plant percent cover nor any of the species’ traits that we measured (Table 1, Appendix A).

## 3. Discussion

This study is one of only a few to document human trampling disturbance in BC’s Coastal Mountains. We focused on the dominant plant species heather (*C. mertensiana*), heath (*P. empetriformis*), blueberry (*V. ovalifolium*), and sedge (*Carex* spp.) on- and off-trails in a popular subalpine and alpine recreation region. We quantified community plant percent cover and measured proxies of plant growth and reproduction. We provide detailed measurements for nearly 1800 individuals of four focal plant species that are of particular regional importance, both culturally, in terms of traditional uses (berries), and ecologically, in terms of their importance within the flora (heath, heather, and sedge meadows). Disturbance reduced plant percent cover and decreased proxy traits for growth and reproduction in some species, although the intensity of these effects varied considerably by species. Unexpectedly, disturbance effects increased with elevation for only *Carex* spp. height, whereas disturbance effects on plant percent cover and species traits stayed constant with elevation. Surprisingly, elevation alone did not have an effect at the community or species level.

### 3.1. Species-Specific Responses

Heather (*C. mertensiana*)*,* heath (*P. empetriformis*), blueberry (*V. ovalifolium*), and sedge (*Carex* spp.) were sensitive to trampling. Human trampling disturbance reduced the size in two out of our four study species (*P. empetriformis, V. ovalifolium)* and reduced reproductive output in just one (*C. mertensiana*). Our empirical findings match those for another study in the nearby Washington Cascades [4], which found reduced height among experimentally trampled, multi-species communities dominated by *P. empetriformis*. Further, their study finding that *Carex* spp. were relatively resilient to trampling disturbance as compared to other taxa corresponds to our finding that *Carex* spp. increased in height with disturbance. Another study in this region found that sedge communities are the most resilient to trampling, and heaths the least resilient [40]. Resilience among sedges and other graminoids may be explained by the presence of adaptations, such as multiple ramets and effective root storage, that buffer against tissue damage [42]. We found *Carex* spp. height, but not diameter, to be sensitive to disturbance impacts, highlighting the need to measure multiple species traits for a better understanding of disturbance effects. While disturbance reduced the diameter of one of our heath species (*P. empetriformis*) and reproductive output in our heather species (*C. mertensiana*), blueberry (*V. ovalifolium*) was most sensitive to disturbance with reduction in both growth proxies (height and diameter).

Other studies have also found differences in disturbance impacts between species and traits [42,45]. Variable responses of flower production to trampling across species might be related to both differences in the sensitivity of flowers to direct trampling and plant response to reduced (i.e., trampled) plant size. Reduced growth from trampling might limit the number of resources available for reproduction, an energetically costly process [46]. Meanwhile, species-specific fruit production responses to disturbance could be related to the effect that disturbance might have on pollinators. For example, insects, which might be negatively affected by human trampling [47], could reduce their pollination services for insect-pollinated species near trails. While it is unclear how negative effects on growth may scale up to influence population persistence in perennial species, decreases in reproduction are more closely tied to plant fitness. However, some work has shown that population performance may be less sensitive to trampling affecting reproduction than survival [19].

### 3.2. No Effect of Elevation on Species Traits

We found no effects of elevation on species traits in our study, and the effects of disturbance only increased with elevation for one species trait. This is surprising, since plant height and reproduction generally decrease with elevation, as higher areas are regionally cooler and less productive [48,49]. The 400 m elevation gradient of our survey may be too small for macroclimatic differences to override microclimatic effects [50]. Moreover, the particular climatic conditions that characterize this maritime-influenced region, particularly the deep, long-lasting snowpack [51], may protect flora against foot traffic in winter. By comparison, in the nearby drier Colorado Rockies there is evidence that vegetation is vulnerable to trampling on windswept and exposed substrates [2]. Snowpack also suppresses tree recruitment, lowering the elevational transition to alpine tundra by several hundred meters in this region relative to less maritime-influenced regions at similar latitudes [51]. As a result, environmental conditions well above the treeline are likely more similar to those at lower elevations in many other alpine zones. In addition, the strong maritime climate moderate conditions here year-round. Moreover, climate change is altering moisture patterns and producing a thinner snowpack here [51,52] and across western North America [53]. Trampling may thus interact with elevation in the future, amplifying the need for further study.

### 3.3. Future Research

Although we did not explicitly investigate species richness, other studies have found that decreased species richness occurred alongside growth and reproductive effects of trampling (e.g., [28]), including in heavily experimentally trampled *P. empetriformis*-dominated communities (e.g., [4]). Our results that disturbance reduces plant percent cover may be in line with previous work that found that disturbance reduces species richness and diversity in an alpine system [28]. Community level impacts from trampling are thus likely, but further study will be needed to evaluate these in the present system, and we are conducting ongoing surveys at our transects to quantify the effect of disturbance on species richness and diversity. Higher levels of species richness have also been associated with higher resilience of subalpine plant communities to trampling [25], making this an important consideration of land management to promote highly mixed rather than single species-dominated vegetation alongside trails.

Another major impact of human disturbance, including trampling, is promoting the survival and spread of invasive species in plant communities by the unintentional introduction of seeds carried on clothes and shoes, as well as the intentional introduction of non-natives for ornamental or other purposes [54,55]. Exacerbating this issue is that non-native species can outperform native species in trampled habitats [54,56,57,58]. However, there is limited data to corroborate such findings for alpine ecosystems. The few studies that have examined disturbance and invasibility find cold-climate ecosystems to be more prone to invasion [59]. While we did not investigate this issue, ongoing studies at our permanent transects examine both changes to species diversity and species invasions.

### 3.4. Implications for Trail Planning

With interest in outdoor recreation in BC rapidly increasing [34,35], there is a need for more trails to spread out disturbance impacts while still providing public access to outdoor spaces. Our findings suggest that some species are more affected by trampling than others and specific management approaches may need to be tailored toward particular species. These varying species-specific impacts should be considered in the planning of future trail and recreation infrastructure (i.e., avoid trails through regions with species that respond strongly to trampling disturbances and/or exist in very localized areas). We suggest that trails through graminoid-, heath- and heather-dominated meadows may be able to recover more quickly from regular disturbance than trails through deciduous shrubs, such as blueberry, based on our findings and those of previous studies [42]. However, it should be considered that sedge meadows often have significantly moister soil and this can result in trail widening (when hikers avoid getting muddy and wet feet) affecting a much larger area and number of individuals (*pers. obs.*). Based on our results that blueberry (*V. ovalifolium*) was most affected by trampling out of the four species we measured, trails should avoid going through these berry ecosystems since it may decrease berry yields for cultural harvesting and for wildlife. We emphasize that our findings only reflect the four species along the elevational gradient that we studied, and future studies are needed on additional species and in other regions to make our findings generalizable.

Although we studied the common species found in well-established, large subalpine meadows, alpine ecosystems are also filled with less common, sometimes endangered, species found in small microhabitats [18]. Based on our findings that individuals on trails were smaller than those off trails, sometimes resulting in less biomass for reproduction, we suggest that trails should avoid going through microhabitats hosting rare or endangered species for whom a decrease in local reproduction could have population-level consequences. Trail shape and infrastructure (e.g., trail markers, boardwalks, etc.) likely also affect community-level sensitivity to trampling. Small and rare communities are likely much more affected by trails that repeatedly go through them, especially if trampling disturbance impacts decrease in a gradient from the trail edge (e.g., switchbacks). Future studies should investigate how far from trail edges plant size and reproduction are affected. If effects are always directly next to trails, then small, rare plant communities should be avoided for any trail development. Finally, our study highlights the need for public education and outreach efforts aimed to minimize trampling damage. There is some ongoing research into how best to achieve this, for example, by providing the public with short courses and learning materials [60], installing educational signs at park trailheads, and erecting deterrent signage and low fencing along trails [61], strategies that have been effective at reducing or spatially confining trampling impacts in US park settings (e.g., Acadia National Park, Maine; Yosemite National Park, California).

### 3.5. Conclusions

Given that we monitored these plant communities and their individuals for only a single growing season, it is notable that we find such reduced vegetative growth and reproduction across species, even if this effect is variable. This suggests a need for long-term studies that include more taxa and explore invasive species as well as species diversity. Human recreation in alpine regions will likely continue to grow, increasing the risk of plant trampling by hikers. Furthermore, continued climate change may further exacerbate species-specific plant vulnerability to trampling. For example, trampled trailing edge populations might be more prone to displacement via colonization from lower elevation species expanding their ranges [62]. Given the decreases in size, reproductive capacity, and plant cover observed in our study, it will be important to evaluate possible interactions between trampling and climate change in other systems where displacement by colonization of lower-elevation species is expected.

## 4. Materials and Methods

### 4.1. Study Site

We conducted this study in the T’ak’t’ak’múy’in tl’a In’inyáxa7n (“Landing Place of the Thunderbird”) region (Garibaldi Provincial Park, British Columbia, Canada) in the Squamish and Lil’wat territories (Figure 3a). The area is classified as Coast Mountain Heather alpine zone within the Biogeoclimatic Ecosystem Classification (BEC) system of BC, where a deep long-lasting winter snowpack limits tree recruitment and the transition to alpine tundra begins at lower elevations than for other alpine zones at comparable latitudes towards the interior [51]. Vegetation is dominated by dwarf shrubs, including *Cassiope* spp. and *Phyllodoce* spp. (mountain heathers and heaths), *Luetkea pectinata* (partridgefoot), and herbaceous species of sedge, grass, and forb. There are patches of stunted trees or “krummholz” at the treeline, comprised mainly of *Abies lasiocarpa* (Subalpine fir), *Picea engelmannii* (Engelmann spruce), and *Tsuga mertensiana* (Mountain hemlock). The mean annual temperatures for the BC alpine have historically been between −4 and 0 °C, with a short growing season and mean annual precipitation of 700–3000 mm, much of which falls as snow [63].

Approximately 100 km from the city of Vancouver, the study area is one of the most popular hiking destinations in southwestern British Columbia [64]. Garibaldi Park attendance has increased substantially, especially since the highway linking Vancouver with the park was upgraded for the 2010 Winter Olympics [33,34,35]. Popular trailheads (Diamond Head, Rubble Creek, and Wedgemont Lake) are filled with hikers and backpackers in the summer season (*pers. obs.*). In the last few years, new trail developments have been proposed (e.g., the Darling Lake trail and the hut developments around the Spearhead Range).

We focused our work on four focal species (*Phyllodoce empetriformis*, *Cassiope mertensiana*, *Vaccinium ovalifolium*, *Carex* spp.) that are common and dominant in the region. We sampled all *Carex* species, both clump forming and rhizomatous. Based on the herbarium records for the area, *C. aenea*, *C. albonigra*, *C. aquatalis*, *C. aurea*, *C. canescens*, *C. circinata*, *C. engelmanni*, *C. hindsii*, *C. illota*, *C. lenticularis*, *C. luzulina*, *C. macrochaeta*, *C. mertensii*, *C. microptera*, *C. nigricans*, *C. pachystachya*, *C. phaeocephala*, *C. preslii*, *C. pyrenaica*, *C. rossii*, and *C. spectabilis* may have been present in the plots.

### 4.2. Data Collection

To answer how hiker trampling disturbance affects common subalpine and alpine plant communities (blueberry, heath, heather, and sedge meadows) along elevational gradients, we chose sites at multiple elevations (1580–1980 m a.s.l.) along three different trails (Figure 3b). We set up 14 permanent 10 m × 0.5 m paired trail-side (disturbed) and off-trail (undisturbed) transects in August 2022 along the heavily used Taylor Meadows-Helm Creek, Black Tusk, and Panorama Ridge trails. We established transects directly adjacent to the trail’s edge and at least a 5 m perpendicular distance away from the trail to compare trampled and untrampled plant communities, respectively (following [28,29]). To mark the trail’s edge, we identified sections of the trails disturbed predominantly by human trampling, but not by water run-off or trail cut construction, thus choosing diffused trail edges with evidence of trampling next to the main trail. We marked the start of each transect with a wooden stake and recorded latitude and longitude in the field with the Gaia mobile application v2022 (WGS 84, ESPG: 4326). This also enabled us to obtain elevation per site (see below).

We recorded plant data in ten continuous 1 m × 0.5 m quadrats within each transect. In each quadrat, we identified all individuals per focal species (*Phyllodoce empetriformis, Cassiope mertensiana, Vaccinium ovalifolium, Carex* spp.) with numbered flags and randomly selected five individuals with a random number generator. If there was a need to eliminate any individuals, it was c. 5 or less. As it is difficult to distinguish ramets from genets of our focal species, we focused on ‘individual’ plants and defined individuals as plants growing out of a central point with all structures growing in the same direction and with a larger spacing to the next plants than within an individual. We recorded the maximum plant height (height perpendicular to the ground without stretching the plant) and the maximum diameter of each selected plant within the quadrat (Figure 3c, Table 2). To keep the measured plant area consistent across all transects, we only measured the structures growing within a given quadrat. To quantify reproductive output of our focal shrub species, we counted the number of buds, flowers, and fruits on each of the five plants randomly selected. We did not count reproductive metrics for *Carex spp.* as it was not possible to accurately distinguish between phenophases. For image analyses of plant percent cover, we took a standardized photograph 1.5 m directly above each quadrat (Figure 3d). We note that while we had mostly *V. ovalifolium* in our transects, at higher elevations there was likely a bit of *V. uliginosum* and *V. deliciosum* and we do not distinguish these in our analyses.

In total, we sampled 1772 plants from 280 quadrats along 14 paired trail-off trail transects across a 400 m elevational gradient between 1580–1980 m a.s.l. These plants comprised 500 *P. empetriformis* individuals, 429 *C. mertensiana* individuals, 634 *V. ovalifolium* individuals, and 209 *Carex* spp. individuals.

### 4.3. Data Processing

We conducted all data processing and analyses in R version 4.2.3 [65]. We used field-recorded latitude and longitude data to obtain elevation per site using the USGS NED1 digital elevation model at 30 m resolution [66]. We used the ‘tidyverse’ [67] and ‘dplyr’ [68] packages to check for unusual patterns or outliers in our data. For an area-standardized reproductive output per individual plant, we first calculated a proxy for plant area by multiplying the maximum plant height by the maximum plant diameter. We then summed bud, flower, and fruit counts and divided this by plant area for an area-standardized summed reproductive output. Finally, we calculated relative reproduction for each individual by dividing summed reproductive output by that species’ maximum reproductive output (i.e., the individual with the highest reproductive output over all transects). While other studies have shown that reproductive output density increases with plant area [69,70], this is not the case in our study in either disturbed or undisturbed transects (tested with a linear mixed model with package ‘lmerTest’; [71]; Appendix A).

To standardize our quadrat photos for percent coverage calculation, we manually flattened and cropped all images to just the quadrat border using Google Drive’s photo editing software (https://photos.google.com). To account for any shading in the plot, we increased image saturation to the point where changes in lighting negligibly affected color intensity. Logs and rocks found in quadrats (25% of photos) were counted as non-green coverage and moss was counted as green cover, although very little moss (8% of photos) was seen in the photos. To account for any shading in the plot and highlight any green vegetation, we increased each image’s saturation by a factor of 30 with a custom algorithm in Python 3.10 [72]. Using the RGB color space, the algorithm then filters the image to only pixels that have a higher green channel digital number (DN) than both red and green DNs, which through testing on a randomly selected subset of images (n = 20) appeared visually to effectively isolate green vegetation (Figure 3d). The fraction of qualifying pixels out of the entire image is then calculated as an estimate of percent plant coverage in the quadrat.

### 4.4. Data Analyses

To test how trail disturbance, elevation, and their interaction affect plant percent cover and species traits, we fit Bayesian generalized non-linear mixed models. We fit separate models for plant percent cover and each species by trait to account for potential response differences. These models allow for non-linear relationships and account for the non-independence of data within a transect pair (i.e., on- and off-trail transect pair) by using a random effect of transect pairs. Our model equation is response ~ disturbance * elevation + (1|transect pair), where ‘response’ is plant percent cover, maximum plant height, or maximum plant diameter. We account for spatial variation among trails using the random effect of transect pair, which accounts for the unique location of the paired transects within a given trail. Quadrats were simply used to facilitate measurements along the entire length of the transect and measurements between quadrats do not vary more than within quadrats, thus we use transect pair as a random effect instead of trail and quadrat.

To account for the left skew in the diameter and height trait data, we fit negative binomial models (link = ‘log’, link_shape = ‘log’) for these response variables and rounded measured diameter and height values to the nearest integer value for this distribution family. The Beta model outperformed the Zero-inflated Beta model (tested on *P. empetriformis*) as checked with a leave-one-out cross-validation (‘loo()’), so to best model our left-skewed proportional relative reproduction data, we fit Beta models (link = ‘log’, link_phi = ‘log’). We also modeled plant percent cover with the Beta distribution. We adjusted 0, 1 values by 0.0001 to keep our data within the Beta distribution interval [0,1]. All models and family distributions are described in Appendix A.

We fit models with the ‘brms’ [73,74,75], ‘rjags’ [76], and ‘R2jags’ [77] packages. We fit models using 3 chains, 5000 iterations and 1000 warmup iterations on 4 cores on macOS Version 12.6.5 with a 2.2 GHz Quad-Core Intel Core i7 processor. We set ‘init = 0’ for plant percent cover and reproduction models. We inferred any parameter estimate’s credible interval that did not contain zero as evidence for an effect of the parameter. For each model, we assessed the model fit by checking that (i) Rhat > 1.1 and that Estimated Sample Size (Bulk_ESS and Tail_ESS) > 1000 (‘summary()’), and (ii) all MCMC chains converged (‘plot()’). We assessed the prior distribution of each model by checking that priors do not overwhelm likelihood (‘powerscale_sensitivity()’). We assessed the posterior distribution by checking that (i) predicted values are similar to posterior distribution with 1000 posterior draws (‘pp_check(ndraws = 1000)’), (ii) scatterplot matrices of posterior parameter distributions are normally distributed (‘pairs()’), (iii) skew is properly modeled using the Fisher-Pearson Skew function (‘ppc_stat()’), (iv) Pareto k values < 0.5 (‘loo()’), and (v) dispersion is properly modeled (‘ppc_loo_pit_overlay()’). The models generally fit well with some exceptions. Skew and dispersion were moderately well to poorly modeled, depending on the species and trait modeled (Appendix A). Some models also had one observation where Pareto K values > 0.7, which slightly decreases the reliability of the Monte Carlo error estimates from Bayesian models. The indicated functions are from the ‘bayesplot’ [78,79], ‘loo’ [80,81], and ‘priorsense’ [82] packages. See Appendix A for additional information on model fit.

## Figures and Tables

**Figure 1 plants-12-03040-f001:**
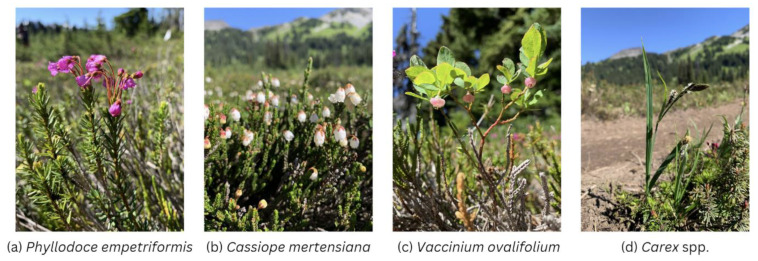
Study species. The four focal species in our studies are common and dominant species of the heath (**a**), heather (**b**), blueberry (**c**), and sedge (**d**) communities in the T’ak’t’ak’múy’in tl’a In’inyáxa7n region (Garibaldi Provincial Park, British Columbia, Canada). They are of cultural importance to the Squamish and Lil’wat Nations and of management concern to the park authority, BC Parks.

**Figure 2 plants-12-03040-f002:**
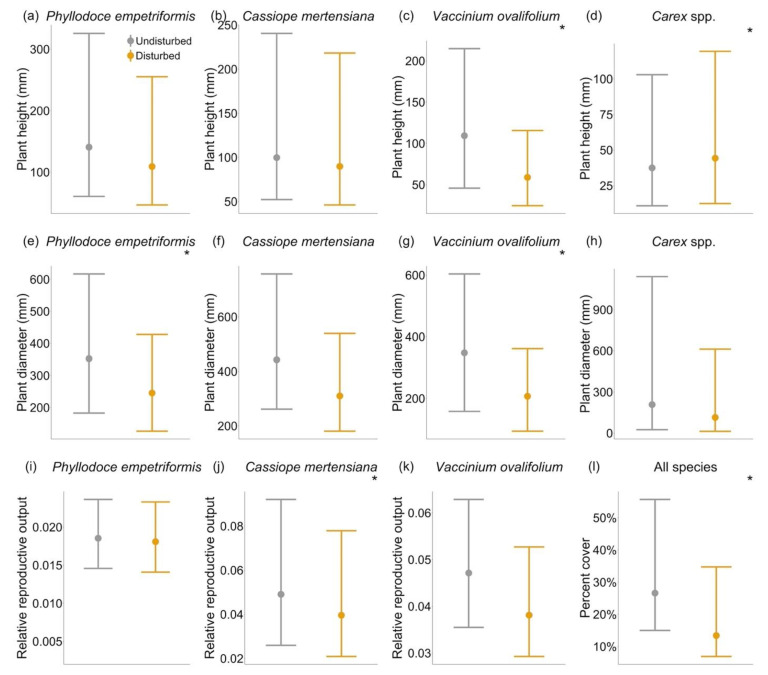
Effects of disturbance on plant traits and plant percent cover. Responses to human trampling disturbance are species-specific, with variable effects on maximum plant height, maximum plant diameter, and relative reproductive output (summed buds, flowers, fruits by total plant area, relative to the maximum per species). Shown are back-transformed parameter estimates (circles) of disturbance with their credible intervals (bars) from Bayesian hierarchical non-linear mixed models. Parameter estimates of disturbance and elevation are shown in Appendix A. Asterisks (*) indicate parameter estimates whose credible intervals do not encompass 0 and can be interpreted as having an effect. Legend for all plots is as in (**a**). Note that y-axes are on different scales.

**Figure 3 plants-12-03040-f003:**
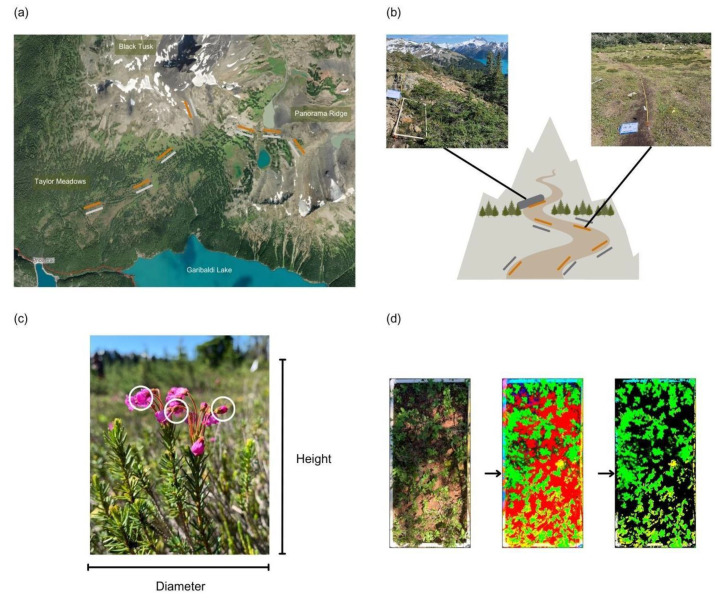
Methods schema. (**a**) We quantified the effects of disturbance along elevational gradients on four dominant subalpine and alpine plant species along disturbed (orange) and undisturbed (grey) transects in the T’ak’t’ak’múy’in tl’a In’inyáxa7n (“Landing Place of the Thunderbird”) region (Garibaldi Provincial Park, British Columbia, Canada) in the Squamish and Lil’wat territories. We established transects (not all shown) along the popular Taylor Meadows, Black Tusk, and Panorama Ridge trails. (**b**) Along each trail (indicated in brown), we established transects (not all shown) at the edge of the trail (orange = disturbed) and 5 m perpendicular to the disturbed transect (grey = undisturbed) below and above the treeline. We recorded plant data in 100 cm by 50 cm quadrats (photo inserts) along each transect. (**c**) At each quadrat, we randomly selected five individuals of each of our focal species to measure maximum plant height and maximum plant diameter (indicated along picture, not to scale), and counted the number of buds, flowers, and fruits (white circles). We did not count reproductive metrics for *Carex* spp. as it was not possible to accurately distinguish between phenophases. (**d**) At each quadrat, we also took a standardized photograph from which to compute plant percent cover.

**Table 1 plants-12-03040-t001:** Estimated parameters of hierarchical models. Sample size (N), intercept, untransformed parameter estimates, and credible intervals (shown in parentheses) from Bayesian generalized non-linear mixed models testing the effects of disturbance, elevation, and their interaction (Dist. * Elev.) on total plant percent cover, and maximum height, maximum diameter, and relative reproductive output of *Phyllodoce empetriformis, Cassiope mertensiana*, *Vaccinium ovalifolium*, and *Carex* spp. Estimated parameters whose 95% credible intervals do not overlap zero are in bold, and can be interpreted as having an effect. Note that we report untransformed estimates from negative binomial and beta distribution models, and in the case of height for *Carex* spp., this means that disturbance increased height (see Figure 2d). We fit models with a random intercept of transect pair and fit separate models for plant percent cover and each species by trait combination to account for different responses. We did not measure reproductive output for *Carex* spp. Additional model fitting information is given in Appendix A.

Species	Trait	N	Intercept	ß_Disturbance_	ß_Elevation_	ß_Dist.*Elev._
[All Plants]	Percent Cover	267	6.58 (0.34, 12.61)	**−8.38 (−12.56, −4.4)**	0 (−0.01, 0)	0 (0, 0.01)
*P. empetriformis*	Height	500	7.04 (2.13, 12.15)	−0.48 (−3.11, 2.12)	0 (0, 0)	0 (0, 0)
	Diameter	500	10.48 (6.66, 14.16)	**−7.15 (−10.53, −3.85)**	0 (0, 0)	0 (0, 0.01)
	Reproduction	500	−4.77 (−7.78, −1.76)	0.39 (−3.47, 4.22)	0 (0, 0)	0 (0, 0)
*C. mertensiana*	Height	429	7.9 (3.74, 12.03)	0.66 (−1.26, 2.55)	0 (0, 0)	0 (0, 0)
	Diameter	429	9.44 (5.9, 12.98)	−2.51 (−5.13, 0.17)	0 (0, 0)	0 (0, 0)
	Reproduction	429	0.66 (−4.18, 5.38)	**−4.28 (−8.13, −0.52)**	0 (0, 0)	0 (0, 0)
*V. ovalifolium*	Height	634	2.13 (−4.7, 8.59)	**−4.58 (−7.28, −1.8)**	0 (0, 0.01)	0 (0, 0)
	Diameter	634	6.39 (−0.05, 12.96)	**−9.18 (−12.34, −5.99)**	0 (0, 0)	0.01 (0, 0.01)
	Reproduction	634	−4.5 (−7.93, −0.73)	−3.76 (−7.99, 0.33)	0 (0, 0)	0 (0, 0)
*Carex* spp.	Height	209	22.62 (5.36, 39.1)	**−19.89 (−30.45, −9.68)**	−0.01 (−0.02, 0)	**0.01 (0.01, 0.02)**
	Diameter	209	−6.29 (−29.7, 16.07)	11.98 (−0.5, 23.86)	0.01 (−0.01, 0.02)	−0.01 (−0.01, 0)

**Table 2 plants-12-03040-t002:** Measured parameters. Description of all the parameters used in this study, and the methodology, unit, spatial scale, and precision of measurement. Precision for plant percent cover is calculated from the pixel size used in our plant percent cover calculation algorithm, and since this varies by camera used, we report the mean pixel size for the lowest resolution camera. Precision for elevation is not available for this region.

Parameter	Method	Unit of Measurement	Spatial Scale	Precision
Maximum plant height	Field measurement	Plant	0.001−1 m	0.001 m
Maximum plant diameter	Field measurement	Plant	0.002–1.5 m	0.001 m
Reproductive Structures (buds + flowers + fruits)	Field counts	Plant	NA	1 count
Plant Percent Cover	Computed from field photos	Quadrat	1 × 0.5 m	0.6 mm^2^ (0.00001%)
Elevation	Computed from field latitude and longitude	Transect	30 m	NA

## Data Availability

The data supporting reported results and R code for analyses can be found on GitHub at https://github.com/ITEX-sites/Garibaldi/tree/main/trampling_analyses (last updated 18 August 2023).

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
