# Peer review of "Species-Specific Responses to Human Trampling Indicate Alpine Plant Size Is More Sensitive than Reproduction to Disturbance"

_plants, 2023, doi:10.3390/plants12173040_

Round 1
Reviewer 1 Report
comments on Chardon et al. Plants ms 2506383
Overall this is an interesting manuscript. There was a tremendous amount
of effort that went into data collection, data analysis, and interpretation.
The results, although not necessarily what the authors expected (no effect
of elevation), still show an impact of trampling in some cases and can help
inform management activity in the Coast Range. On one hand someone may
argue "of course there is higher impact along a trail than off trail"
but the differences among species help us understand the nuance and
relative responses of these species, which then helps us understand how
to best manage for these impacts.
This is a good, thorough study that should be published for other
researchers to learn from. In study design, perhaps the biggest question
is how the authors placed the 'disturbed' quadrats because trail edges
can be pretty abrupt. This could be addressed more thoroughly. In the
study analysis, please see the discussion below about including more
random effects in your design or otherwise accounting for the nested
replication.
The introduction is great and sets the stage well.
Specific comments follow.
Introduction
P1. Not sure the sentence about Cassiope is correct (or relevant?) as the cited report
discusses long-abandoned disturbance and succession dynamics.
Pg2 Para3: US and Canada preferred spelling seems to be emphasize, not emphasise
Pg2 Paragraph beginning with "To our knowledge". You flipped citation styles.
Newman & Powell and Nepal & Way are not in the Lit cited.
Pg2 final paragraph. Replace "Our findings will.." with "Our goal is to.."
Pg3, top. Remove hyphens after "on" and "off". (too distant from
hyphenated word 'trail' and can stand alone.)
Pg3, bottom. You note that the graminoids have multiple ramets, but the other
three species are also clump forming and often prostrate. See:
http://floranorthamerica.org/Phyllodoce_empetriformis
http://floranorthamerica.org/Cassiope_mertensiana
http://floranorthamerica.org/Vaccinium_ovalifolium
in all cases, FNA suggests these are low, clump forming shrubs. In my
experience it is difficult to differentiate ramets from genets
with alpine shrubs like this. This question is relevant for your
measurements as well. If it is difficult to tell whether a shrub is an
individual genet, then how do you measure height and width? You
shoud discuss how you made these decisions in the methods.
Pg 4, top. You get into the results in the final paragraph of the
introduction. Everything after the first sentence ("We used Bayesian..")
should probably be removed.
Methods and Materials
Pg 9, Study Site. You mention the character of the shrubs here
("dwarf shrubs"), and this could be a good place to address possible
measurement issues/decisions. Or add another paragraph describing your
study species.
You also need to describe the Carex species you included with a little
more detail. The genus Carex is huge with forms ranging from extreme clump-forming
to strongly rhizomatous mat-forming species. Even if you didn't identify
to species (which is fine) you need to describe the species expected
in these habitats, which forms you included, and their typical growth forms.
And how did you distinguish genets?
Pg 10. How were the transects set up as permanent? Iron spikes in rock/soil?
GPS only? Transect ends or for each quadrat?
pg 11, top. How did you randomly choose each of the five individuals?
What was the typical number of individuals in a quadrat these were chosen
from (6? 16? 60?) Simply assure us you minimized bias here.
pg 11. "..maximum diameter of plant within quadrat." So, does that mean that
if a 0.75 meter diameter clump extends only 0.3 m into the quadrat, it
gets a value of 0.3? (possibly looks like that scenario in quadrat pictured
in Fig 3b, top left photo, top right of quadrat). you should justify
the reasoning for this.
Pg 11. Table 2. Did you really measure height down to 0.1 mm? And how
do count data have a scale ranging from 0.1mm to 1.5m? I'd probably change
that column header to 'scale' (or better, 'precision') so you can tell us
the precision used for percent cover (down to an even 10%, 1% 0.1%?).
And elevation is 30m pixels, but what's precision for elevations? (1m?, 0.1m?)
Pg 11. "...by dividing summed reproductive output by that species’ maximum
reproductive output (i.e. individual with highest reproductive output)"
Was that the individual with highest output in the quadrat, transect, trail,
or over all trails/all samples?
Pg 11. On the discussion of reproductive efficiency, where you cite Fig S2.
My read of your citation 65 seems to suggest the same pattern as you (in
their last paragraph):
"reproductive efficiency tends to be greatest in smaller and early reproducing
individuals"
Slight tangent but Figure S2 seems to be showing a clear 'boundary effect'
where where points could fall anywhere below a curve (poisson? negative binomial?)
and that curve could be estimated with quantile regression. To understand
mechanism would be a study in itself.
Pg 11. parag before data analysis. you probably should post the python
code as a supplement (or on github, e.g.) to make the work repeatable.
For example, was there a training phase to determine the threshold
for 'greenness'? Was this repeated for all images or assumed to be
consistent across all images?
Pg 12, top, where you show the model (thank you for showing!). Your design
is nested like this:
3 trails, 14 permanent 10m transects * 10 quadrats (transect
pairs) * 5 individuals
yet you only account for "transect pair" as a random effect in the model.
It isn't clear if this is transects or quadrats. If transects, classically
that could mean that the individuals were subsamples within quadrats
(and thus averaged by quadrat) and quadrats were subsamples within
transect (and thus averaged by transect). So your sample size was 14. If
quadrats, then were samples averaged within quadrats?
I assume neither of those are the case - you'll have to explain how you accounted
for the 'pseudo-replication' here. (Possibly the model you used is just
the best fit?). why not add random effects for trail, transect?
Results.
Pg 4. last sentence. Replace 'grasses' with 'graminoids' or 'sedges'
Pg 5. Table 1. I like this format and the information provided.
Discussion.
Pg 7. In the methods you indicate you sampled 5 individuals of each species
in every quadrat. But here, you suggest these are representative of their
own communities (first sentence of Species-specific responses section).
Which implies when you find one (or one group: heath,
blueberry, sedge) you are unlikely to find the others. Which is it?
You should probably report statistics of frequency of each species
in the plots. That actually might be a productive analysis. Could you
get five individuals in all quadrats? if not, what's the pattern?
Pg 7, second paragraph from bottom: "Reduced growth from trampling..."
The problem with the discussion here is that you found higher
reproductive density in smaller plants (Fig S2). If you keep this
part of the discussion you'd have to account for that.
Page 8. Bottom. Along with trail infrastructure you should discuss outreach
and education, which can have a big impact. Certainly a different
environment, but there's good success in the northeast US with education
and outreach.
Reviewer 2 Report
The authors researched the effect of trampling on alpine/subalpine ecosystems, and how it changed along elevational gradients. They found that trampling could reduce plant cover and impact all species, but that effects vary by species and traits. They also found that the trampling effects only varied with elevation for one species’ trait, and growth traits were more sensitive to trampling than reproductive traits. This research can help us minimize he effect of trampling on alpine/subalpine ecosystems. However, there are still some problems that need to be addressed by the authors to improve the quality of the article.
1) The format of this article is not standardized and there is no page number and line number. This has caused us some trouble in reviewing the manuscript.
2) I think some paragraph were placed in the wrong section. For example, the following paragraph “We used Bayesian hierarchical models to test for the effects of disturbance, elevation, and their interaction on plant percent cover and species traits … or can be generalized for broader groups (e.g. other graminoids, evergreen shrubs, or deciduous shrubs [40])” should be deleted from the Introduction. “In total, we sampled 1772 plants from 280 quadrats along 14 paired trail-off trail … which slightly decreases the reliability of the Monte Carlo error estimates from Bayesian models.” Should be removed to the Materials and Methods section.
3) According to the experimental design I think Paired t test should be used to analyze the difference between plant percent cover of on trail transect and that of off-trail transect.
4) Why the authors used no-linear models instead of linear models in analyzing the effect of trampling, elevation, and their interaction on species traits?
5) The table in a scientific paper should be a three-line table
6) What are the numbers in parentheses and the numbers outside parentheses in Table 1?
Round 2
Reviewer 2 Report
None.